# The Importance of Intestinal Microbiota and Dysbiosis in the Context of the Development of Intestinal Lymphoma in Dogs and Cats

**DOI:** 10.3390/cancers16122255

**Published:** 2024-06-18

**Authors:** Wioleta Jadwiga Breczko, Joanna Bubak, Marta Miszczak

**Affiliations:** 1EZA Student Science Club, The Faculty of Veterinary Medicine in Wroclaw, Wrocław University of Environmental and Life Sciences, 31 Norwida St., 50-375 Wrocław, Poland; 2Department of Pathology, Division of Pathomorphology and Veterinary Forensics, The Faculty of Veterinary Medicine in Wrocław, Wrocław University of Environmental and Life Sciences, 31 Norwida St., 50-375 Wrocław, Poland; joanna.bubak@upwr.edu.pl; 3Department of Epizootiology and Clinic of Birds and Exotic Animals, Division of Infectious Diseases and Veterinary Administration, The Faculty of Veterinary Medicine in Wroclaw, Wrocław University of Environmental and Life Sciences, Grunwaldzki Sq. 45, 50-366 Wrocław, Poland; marta.miszczak@upwr.edu.pl

**Keywords:** lymphoma, intestines, microbiota

## Abstract

**Simple Summary:**

Cancer of the lymphatic system is a prevalent disease in dogs and cats, with older pets being the most affected group. However, it is crucial to recognize that animals of any age can develop this condition. Intestinal lymphoma, a type of neoplasm, arises when lymphocytes—white blood cells—become cancerous and proliferate uncontrollably. Recent research suggests that the microbiota, the community of bacteria and other microorganisms in the intestines, may play a role in the development of various diseases, including intestinal lymphoma. A healthy microbiota supports the immune system and prevents harmful cellular changes. In contrast, an imbalance in these microorganisms, potentially caused by poor diet, antibiotic use, or illness, can lead to inflammation and increase the risk of cancer.

**Abstract:**

Recent advancements have significantly enhanced our understanding of the crucial role animal microbiomes play in veterinary medicine. Their importance in the complex intestinal environment spans immune modulation, metabolic homeostasis, and the pathogenesis of chronic diseases. Dysbiosis, a microbial imbalance, can lead to a range of diseases affecting both individual organs and the entire organism. Microbial disruption triggers inflammatory responses in the intestinal mucosa and disturbs immune homeostasis, increasing susceptibility to toxins and their metabolites. These dynamics contribute to the development of intestinal lymphoma, necessitating rigorous investigation into the role of microbiota in tumorigenesis. The principles explored in this study extend beyond veterinary medicine to encompass broader human health concerns. There are remarkable parallels between the subtypes of lymphoproliferative disorders in animals and humans, particularly Hodgkin’s lymphoma and non-Hodgkin’s lymphoma. Understanding the etiology of a cancer of the lymphatic system formation is critical for developing both preventive strategies and therapeutic interventions, with the potential to significantly improve patient outcomes. The aim of this study is to discuss the optimal composition of the microbiome in dogs and cats and the potential alterations in the microbiota during the development of intestinal lesions, particularly intestinal lymphoma. Molecular and cellular analyses are also incorporated to detect inflammatory changes and carcinogenesis. A review of the literature on the connections between the gut microbiome and the development of lymphomas in dogs and cats is presented, along with potential diagnostic approaches for these cancers.

## 1. Introduction

### 1.1. Background

Primary intestinal lymphomas are a group of malignant lymphoproliferative tumors and represent the most common form of this type of transformation in cats, accounting for approximately 55% of all intestinal tumors [1,2,3]. In dogs, these lymphomas are diagnosed less frequently, constituting about 8% of all intestinal tumors, yet they still represent a significant proportion of lymphoma cases [1,3,4]. The prevalence of intestinal lymphoproliferative disorders in veterinary patients presents a significant challenge. Identifying the causes and conducting further research into the mechanisms of carcinogenesis is crucial for developing effective strategies for diagnosis and treatment. Recent studies suggest that alterations in the microbiota may significantly contribute to the development of chronic inflammatory processes, immune system alterations, and the occurrence of cancer, including intestinal lymphomas (Figure 1) [5,6,7,8].

### 1.2. Objectives of the Article

The objective of the article is to present the current state of knowledge regarding the relationship between intestinal microbiota and the development of intestinal lymphoma. Understanding this connection could provide opportunities to reduce the risk of intestinal lymphoproliferative disorders by restoring the proper composition of the microbial ecosystem and enhancing functionalities, for example, through the use of probiotics and prebiotics. At present, the only confirmed factors contributing to the occurrence of intestinal lymphoma in cats are the carriage of feline leukemia virus (FeLV) and feline immunodeficiency virus (FIV) in cats [3].

## 2. The Microbiota

### 2.1. Definition and Composition

The primary function of the digestive system is to supply the body with nutrients by absorbing them [7,10]. The microbiota plays a crucial role in several biological processes, including nutrition immunological, metabolic, and energy functions. This group of microorganisms with the correct qualitative and quantitative composition is a crucial component of the body’s immune mechanisms and is also involved in the production of vitamin B_12_ and other metabolic processes [6,11,12,13,14,15]. The collection of aerobic and anaerobic microorganisms residing in the intestines and the surrounding environment is referred to as the microbial population [10,11,12,16]. Conversely, the microbiome evolves with the host organism and encompasses the genetic content of the microbiota [13,17]. The relationships between the microbiota and the host organism are crucial [18]. Close genetic, metabolic, and immunological interactions between them produce effects both locally and throughout the body [5,19]. These interactions help maintain a balance between the body’s tolerance and activation for protection [20]. The microbiota consists of bacteria (the largest group), archaea, fungi, protozoa, and viruses. The physiological composition of microorganisms in dogs and cats is presented in Table 1.

### 2.2. Functions of the Microbiota

Animals are born with a sterile digestive tract that becomes colonized a few hours after birth [11]. The types of microorganisms colonizing the intestines of newborns are influenced by the composition of the microbiota in the mother’s birth canal and the surrounding environment, as well as maternal antibodies delivered in colostrum [11,25]. It is important to note that the presence of microbial communities allows the full maturation of the gut-associated lymphoid tissue (GALT). GALT, which is associated with the mucous membranes of the digestive tract, matures comprehensively, integrating components of both cellular and humoral responses [25].

The immune barrier’s previously mentioned role involves collaboration between the body and intestinal microbial society. Commensal microorganisms assist in maintaining the body’s homeostasis; conversely, their antigens are tolerated by the host’s immune system. Additionally, the microbiota also plays a role in priming the immune system to recognize potential threats [7]. It may also support the formation of helper T lymphocytes (Th17) due to the presence of segmented filamentous bacteria (SFB) of the genus *Clostridium*, thereby aiding in maintaining bodily stability. However, it may also contribute to the occurrence of both local and systemic inflammation. The mutual regulation between the microbiotic environment of the gut and the immune system is facilitated by enabling the IgA reaction, which influences the composition and function of the intestinal microbiota. In individuals with a genetic predisposition, indirectly related to IgA deficiency in the intestine, the risk of disease escalates due to an increase in the content of anaerobic microorganisms and hyperplasia of isolated lymphoid follicles (ILFs) [5,6]. Mucus also plays a crucial role by separating the intestinal mucosa from the microbiota, thereby preventing its displacement [7]. Both barriers make it challenging for microorganisms to breach the epithelial cells from the intestinal lumen [26,27]. An important factor influencing changes in the microbial communities and, consequently, the functioning of the immune system is diet, to which it is highly sensitive [7].

## 3. Normobiosis and Dysbiosis

### 3.1. Gut Microbiota and Lymphoma

The association between bacteria and cancer is not a recent discovery. In the 19th century, research in human medicine demonstrated the anti-cancer effect of bacteria [28]. The gastrointestinal tracts of cats and dogs host a complex community of microorganisms—including bacteria, archaea, fungi, protozoa and viruses, which are essential for the host’s health and various biological functions. The gut microbiota is the most plentiful and varied microbial community found in various anatomical compartments. It encompasses in excess of 1000 bacterial species, making it the most diverse of all microbial communities [29]. Due to the considerable diversity of the bacterial biota and its intricate interactions with the intestinal epithelium, the examination of the biological community of intestines poses challenges [26]. Recent studies have demonstrated that the microbiotic environment of the gut and its functions in dogs and cats are similar to those in humans [30]. Short-chain fatty acids (SCFAs) produced by these microbes are indispensable for energizing the colon lining cells and supporting microbial metabolic functions. Furthermore, the microbiota plays a pivotal role in regulating bowel movements and orchestrating the synthesis of anti-inflammatory compounds, thereby fostering holistic gut health [29,31]. In a pertinent experiment conducted by Furusawa et al. (2013) involving mice, it was demonstrated that it possesses the capability to induce regulatory T lymphocytes (Treg) through the overproduction of butyrate. Similar observations have been documented in humans [27].

### 3.2. Normobiosis

Normobiosis represents a state in which the body’s bacteria are in balance, carrying out specific functions (Figure 2A) [10,11,32]. These microorganisms exhibit an antagonistic effect on harmful counterparts, due to their resistance to colonization [11]. Research indicates that the microbiotas of dogs and cats have grown more diverse through domestication, resembling that of humans [1,22,33]. While microorganisms naturally present in the microbiota typically exist in small numbers, they can lead to diseases; these are termed pathobionts. An example is *Enterobacteriaceae*, which increases not only during infections but also in other inflammatory conditions [34].

### 3.3. Dysbiosis

As individuals age and experience changes in diet, environmental factors, and, notably, diseases, modifications occur in the intestinal microbiota, affecting the transcriptome, proteome, and metabolome [10,40]. The imbalance is further influenced by factors linked to societal progress, which may be of significance for animals, such as alterations in diet, parasite elimination, and antibiotic overuse [7]. In conditions of disrupted intestinal microbiota composition (dysbiosis), functions and activity transform, leading to metabolic changes, including the transformation of tryptophan, short-chain acids, and secondary bile acids with immunomodulatory functions [15,41]. This directly correlates with impaired immune system functioning [7]. The occurrence of intestinal dysbiosis is based on three factors: a decrease in bacterial diversity, a reduction in the stability of bacterial populations, and a decrease in the occurrence of anaerobic bacteria—particularly from the *Firmicutes* and *Bacteroidetes* families—which favors facultative anaerobic bacteria from the *Enterobacteriaceae* family [35]. These changes may be induced by either a disease or the treatment of another disorder [34]. Additionally, individual cell populations induce the synthesis and secretion of metabolites, cytokines, and hormones [7].

Dysbiosis and intestinal inflammation may play a role in the development of gastrointestinal lymphoma (Figure 2B). Studies have demonstrated that 33% of lymphocytic cancer originating from large granular lymphocytes of the gastrointestinal tract and 60% originating from T lymphocytes occurred concurrently with chronic inflammation in the affected area [36]. The presence of bacteria contributes to the development of chronic inflammation, which may eventually lead to carcinogenic processes [6,7]. Inflammatory reactions associated with inflammatory bowel disease (IBD) may also serve as a foundation for the development of lymphoma [42]. The mechanisms underlying carcinogenesis in the context of inflammation remain incompletely understood. Key factors in this process include the production of bacterial toxins, the release of molecular danger signals, and alterations in the intestinal microbiota. Additionally, disruption of intestinal barrier function and suppression of anti-inflammatory mediators play crucial roles. Advanced stages of abnormal cell development further contribute to cell damage and chromosomal disorders, highlighting the multifaceted nature of inflammation-driven carcinogenesis [23]. In the current investigation, an elevation in the level of Treg lymphocytes was identified in intestinal tumor lesions in half of the lymphoma group (four individuals). Additionally, an increase in Gram-negative facultative anaerobes, specifically *Parabacteroides*, was observed (refer to Figure 2C) [27].

## 4. Lymphomas

Initial evidence linking malignant tumor occurrence to microbiota composition emerged from human gastric cancer cases, associating tumor development with the presence of *Helicobacter pylori* [6,7]. Nevertheless, a direct relationship between this bacterium and the development of lymphoproliferative disorder has not been clearly proven in animals. Although one case of lymphoma was reported to be associated with *H. pylori*, the correlation remains unconfirmed [3].

### 4.1. General Information

Cancers of the lymphatic system are a diverse group of malignancies that can manifest in any site containing lymphatic tissue [43]. In the digestive system, lymphomas most frequently develop in the jejunum and ileum [36,44]. Originating from the lymphoreticular system, they involve both T lymphocytes (typically in the small intestine) and B lymphocytes (commonly in the lymph nodes and stomach) [27,45,46,47,48]. These cells, functioning as defensive cells associated with the mucous membrane, are continually exposed to contact with foreign, potentially harmful bacteria. Consequently, they must consistently receive signals from dendritic cells and antigen-presenting cells (APC), leading to inflammation and the potential for cancer, particularly in the colon and stomach [20].

Lymphomas can be classified based on their location, with nodal occurrences in peripheral lymph nodes, mediastinal locations, and extranodal appearances in the gastrointestinal tract, kidneys, or nervous system [49,50]. The multifocal form is characterized by the presence of enlarged, painless peripheral lymph nodes [43]. Intestinal (gastrointestinal) lymphomas may either occur as primary lesions or infiltrate the intestines and stomach [36,47]. Regarding malignancy, they are categorized into three grades: low, intermediate, and high, with a distinct type—specifically related to the gastrointestinal tract—known as large granular lymphocytic lymphoma [36].

Typically, the clinical signs of cancer of the lymphatic system are nonspecific and vary depending on the cancer’s location [1,48]. In the digestive form, these often include anorexia, vomiting, diarrhea, malabsorption syndromes, weight loss, tarry stools, and the presence of fresh blood in the stool, as well as acute, subacute, or chronic weakness and lethargy [47,48,51,52].

### 4.2. Lymphoma in Cats and Dogs

In terms of specific tumor types, lymphoma accounts for nearly 30% of all feline tumors, making it the most common cancer, located primarily in the gastrointestinal tract in most studies, and the number of cases is increasing [1,2,3,36,49,51]. Currently, the median age of onset is 10–12 years [24]. Cats belonging to the following breeds are predisposed to developing intestinal lymphoma: domestic short-haired cats, domestic long-haired cats, Siamese, Persians, and Orientals [3,47,48]. Intestinal lymphoma originates from both B and T lymphocytes—equally or with a predominance of the latter [3]. Most often, these are lesions of intermediate or high malignancy, primarily located in the jejunum. Additionally lymphoproliferative disorders with a low degree of malignancy are observed, typically present along the entire length of the intestine [2,35,46,53]. In both cases, the mesenteric lymph nodes are involved in the cancer process [2,46]. The diagnosis of cancer of the lymphatic system is typically made in stage II or III of advancement [1].

In dogs, intestinal lymphomas represent the second most common intestinal neoplasm (after adenocarcinoma), with a predilection for the small intestine [1,3,51]. The incidence is estimated to be 5–7% for all types of lymphatic cancers [46,54]. The median diagnosis of pathological changes occurs at the age of 6–9 years [27]. Shar-Peis and Boxers show a breed predisposition [54]. They typically exhibit a high or intermediate degree of histological malignancy [27].

The distinction between IBD and small intestinal lymphoma in cats is often difficult to make, particularly when only endoscopic biopsy specimens are available for evaluation. In a study conducted from 2009 to 2012, Omori and co-authors demonstrated differences in the bacterial biota between healthy and sick animals. The study included 31 dogs, comprising 15 healthy dogs and 16 diagnosed with IBD or intestinal lymphoma. The microbiota composition of veterinary patients with intestinal lymphoma veterinary patients differed significantly from that of healthy or IBD dogs. In animals with malignant tumors, there was an increase in the occurrence of microorganisms from the *Eubacteriaceae* family, including *Eubacterium hallii*, which is a Gram-positive bacterium [27,33]. However, it should be noted that the mere presence of IBD may lead to cancer. The presence of certain bacteria, such as *Helicobacter pylori*, *Enterobacteriaceae* or *Fusobacterium* spp., can result in the development of inflammation. Consequently, the intestinal mucosa can be infiltrated by effector cells, which creates an environment conducive to tumor formation [23].

Garraway et al. (2018) presented findings indicating an increase in *Fusobacterium* and *Bacteroides* in cats with intestinal lymphoma compared to those with IBD [23]. Similarly, Marsilio et al. (2019) observed a trend towards microbiota alterations in these cats, though these changes did not reach statistical significance after adjustment [35]. Sung et al. (2022) conducted a comprehensive study on the microflora composition in cats with chronic intestinal diseases, demonstrating significant reductions in *Bacteroides*, *Bifidobacterium*, *C. hiranonis*, *Faecalibacterium*, and *Turicibacter* and increases in *E. coli* and *Streptococcus*. These findings suggest that chronic intestinal conditions in cats are associated with notable shifts in microbial populations [31]. Other studies have also observed significant changes in the microbiotas of dogs and cats with various intestinal conditions, as outlined in Table 2.

The methods of differential diagnosis between IBD and lymphoma are presented in Table 3 and Table 4. The use of ultrasound does not allow for a clear differentiation of diseases. In both cases, the intestinal wall is thickened (above the reference range of 0.16–0.36 cm), and changes may occur in individual layers of the organ wall. The images may also show lymph node enlargement [23,55]. It is also possible that no changes may be observed, but this is more frequently the case in IBD [47].

**Table 2 cancers-16-02255-t002:** A review of the research literature on the relationship between the gut microbiota and lymphoma.

Authors	Research	Date	Animals	Methodology	Results
Omori et al. [27]	Fecal microbiome in dogs with inflammatory bowel disease and intestinal lymphoma	2009–2012	dogs	Fecal samples were collected from 3 groups of dogs: 11 healthy dogs, 16 dogs diagnosed with IBD, 7 dogs diagnosed with intestinal lymphoma, 15 dogs with no clinical signs.	- Increase in *Eubacteriaceae* and *Parabacteroides* spp. in intestinal lymphoma.
Garraway et al. [23]	Relationship between the mucosal microbiota and gastrointestinal inflammation and small-cell intestinal lymphoma in cats	2018	cats	Tissue samples were collected from 14 cats diagnosed with IBD and 14 cats diagnosed with small-cell GI lymphoma (SCIL). No healthy cats were included as controls. The procedure involved biopsy sampling of the intestine; the samples were subsequently evaluated for the presence of bacteria, NF-κB transcription factor expression, and CD11b+ cells.	- Increased levels of *Fusobacterium* spp. and *Bacteroides* spp. are observed in cats with SCIL. - Further research is needed to determine whether this is a cause or an effect of the disease.
Marsilio et al. [35]	Characterization of the fecal microbiome in cats with inflammatory bowel disease or alimentary small-cell lymphoma	2019	cats	Fecal samples were collected from 38 healthy cats, 13 cats diagnosed with IBD, and 14 cats diagnosed with small-cell lymphoma.	- Observed a tendency towards changes in bacterial composition—after statistical correction, they were not significant. - Increased levels of *Firmicutes*, *Bacteroides*, and *Actinobacteria*, as well as the families *Enterobacteriaceae* and *Streptococcaceae*.
Mahiddine et al. [56]	Microbiome Profile of Dogs with Stage-IV Multicentric Lymphoma: A Pilot Study	2022	dogs	Fecal samples were collected from 11 healthy dogs and 7 dogs diagnosed with lymphoma.	- The composition of the gut microbiota varies based on an individual’s health status. - Stage-IV multicentric lymphoma exerts a distinct influence on the gut microbiome. - There are significant differences in the populations of *Actinobacteria* and *Bacteroides* between sick and healthy dogs. - An increase in *Streptococcus lutetiensis* and *Corynebacterium amycolatum* has been observed.
Sung et al. [31]	Dysbiosis index to evaluate the fecal microbiota in healthy cats and cats with chronic enteropathies	2022	cats	Fecal samples were collected from 80 healthy cats and 68 cats diagnosed with chronic enteropathies, which included both IBD and alimentary small-cell lymphoma.	- Reduction of the quantities of Bacteroides, Bifidobacterium, Clostridium hiranonis, Faecalibacterium, Turicibacter in cats with chronic enteropathies. - Increased *E. coli* and Streptococcus levels in cats with chronic enteropathies.

**Table 3 cancers-16-02255-t003:** Currently available studies for differential diagnosis in cases of intestinal inflammation/diarrhea based on history, physical examination, and general blood examination [48,57,58].

Diseases that May Cause Similar Clinical Signs in the Intestinal Area	Type of Examination in Differential Diagnosis
Food allergy/sensitivity	Elimination diet containing protein hydrolysates/protein from a new source (possibly allergy tests)
Parasites	Stool flotation test, smear, SNAP test (e.g., *Giardia* spp.)
Bacterial intestinal inflammation	Microbiological examination of stools—smear or in the laboratory
Metabolic or systemic diseases	Biochemical blood test with a general profile
Exocrine pancreatic insufficiency	Trypsin-like factor immunoreactivity (TLI)
Acute pancreatitis	Pancreatic lipase (fPLI or cPLI), imaging diagnostics
Hyperthyroidism (mainly in cats)	T_3_, T_4_
Vitamin B12 deficiency	Cobalamin
Addison’s disease	Cortisol—ACTH stimulation test
Gastrointestinal obstruction	Diagnostic imaging
IBD	Endoscopic biopsy, laparotomy with biopsy, laparoscopy with biopsy, cobalamin
Cancers (Adenocarcinoma, Lymphoma, others)	Diagnostic imaging, endoscopic biopsy, laparotomy with biopsy, laparoscopy with biopsy
Fungal infection	Microbiological examination of stools, endoscopic biopsy, laparotomy with biopsy, laparoscopy with biopsy

**Table 4 cancers-16-02255-t004:** Tests for molecular and cellular analysis of IBD and lymphoma.

Type of Examination	Cytological Examination	Histopathological Examination	Immunohistochemical Examination (IHC)	PCR Examination for Rearrangement of the Antigen Receptor (PARR)	Flow Cytometry (FC)
Analysis					
Site of sample collection	Altered lymph nodes/tissue fragments [47,52,59]	Infiltrative changes + enlarged mesenteric lymph nodes [47]	Infiltrative changes/transformed tissue samples/lymph nodes [59]	Altered lymph nodes and tissues [60]	Blood, lymph nodes, tonsils, transformed tissue fragments [61]
Material	Fine-needle aspirates [47,52]	Biopsy specimens (not smaller than 1 mm) or entire lesion—avoid areas of necrosis and ulceration; fixed in formalin [52].	Fresh samples—fine-needle aspirates and biopsies—or embedded in paraffin [60]	Fresh samples (fine-needle aspirates, biopsies, blood, body fluids) or fixed in formalin or made from a paraffin block [47,60]	Whole blood collected in tubes with sodium EDTA, tissue material—fine-needle aspirates—cells isolated in a density gradient [61,62,63]
Differentiation between lymphoma and IBD	Difficult for low-grade lymphomas (LG low grade); possible for large cell lymphomas (DLBCL) and lymphomas with granular lymphocytes [47]	Possible, but the test result may be unreliable [47,62]	Possible [47,62,64]	Difficult for low-grade lymphomas (LG low grade); possible for large cell lymphomas (DLBCL) and lymphomas with granular lymphocytes [42,47,62]	Possible [62]
Intestinal lymphoma	Small lymphocytes with few blasts [47]	Small/large lymphocytes—the possibility of assessment depends on the place where the biopsy sample was taken, lymphoblasts [47,65]	CD20+ for B lymphocytes/CD3+ for T lymphocytes [60,64,66]	Expression of heavy-chain immunoglobulin (IgH+)/T-cell receptor gamma (TCRγ+) [60]	Identification of markers for different types of lymphoma [61]
IBD	Inflammatory cells, including plasma cells, numerous blasts [47]	Inflammatory cells, including plasma cells [57]	Differentiated immunophenotype [67]	Negative [47]	Immunophenotyping—TCRγδ+ T-cell receptor (decreased), CD21+ for B lymphocytes (decreased) [67]
Diagnostic value	Moderate, additional confirmation for the presence of proliferation is necessary [47]	High—higher with appropriate cooperation between the referring physician and pathologist [52]	Moderate, false + or − results may occur. However, they allow adjusting the likely course of the disease and treatment plan (e.g., by determining genetic changes and the degree of proliferation) [47]	High, especially used to differentiate lymphomas from reactive proliferation [47,60]	The use of flow cytometry (allowing qualitative and quantitative analysis of cells and antigens) along with fine-needle aspiration biopsy increases the accuracy in detecting lymphoma, along with its subclassification [61,62,68]
Notes	Inability to assess the structure of a specific tissue, assessment of cells/individual clusters only [47]	Abandonment in cases where the result has no impact on the treatment method or when the collection is too dangerous [52]	For better diagnostics, it is advisable to complement with PARR examination and monitor treatment effects [47]	Diagnosis of the presence of monoclonal lymphocyte proliferation; additional HP/IHC diagnostics required; consider the possibility of result distortion due to monoclonality or prior corticosteroid administration [47].	The requirement is to provide live cells for the study [61]

## 5. Diagnostics

### 5.1. Clinical Signs

Clinical signs accompanying the presence of intestinal lymphomas may resemble the ingestion of a foreign body, intestinal ulceration, dilatation of lymphatic vessels, or inflammation on an identified background, and intussusception. Other types of gastrointestinal cancer should also be excluded [48].

### 5.2. Diagnostic Procedures

In terms of clinical manifestations, the diagnosis of cancer of the lymphatic system may include anemia, hyperleukocytosis, and the presence of lymphoblasts in peripheral blood, increased liver enzyme activity (in the case of liver involvement), and monoclonal gammopathy [48].

The palpability of the tumor mass during abdominal examination in intestinal lymphoma varies by species. In dogs, palpation is generally not a reliable method for detecting the tumor (with a positive result in 20–40% of cases). In cats, palpation is a more effective diagnostic tool, with an estimated 86% success rate in affected animals [3]. The reference standard for diagnosing this disease is a biopsy of the intestinal wall. While surgical biopsy is the preferred method, endoscopic biopsy is also a viable option [2]. The former method, which involves obtaining a full cross-section of the intestine, proves more effective in differentiating T-cell lymphoma from lymphocytic–plasmacytic enteritis in cats, an example of IBD. Subsequently, the diagnosis, based on the examination of the biopsy specimen, typically follows WSAVA or WHO guidelines [69].

In addition to biopsy, peritoneal exudative fluid may be collected if present. It is noteworthy that the clinical signs and treatment for inflammatory bowel disease and low-grade lymphomas are similar [47,53]. Consequently, differential diagnosis is occasionally refrained from due to cost considerations [2,35]. Furthermore, attention is also given to the co-occurrence of inflammation in lymphoma as a secondary reaction or in response to the developing cancer [69].

The current research in the field of treatment and prophylaxis is focused on modifying the microbiota in order to enhance the digestion and absorption of nutrients, thereby supporting the treatment of diseases affecting the gastrointestinal system [10]. The test for identifying bacteria in the intestines typically requires the collection of fresh feces directly from the anus of animals or through a swab from the rectum [27].

The most straightforward method for obtaining material is as follows. Moreover, ethical considerations are crucial, particularly regarding the collection of samples from other parts of the intestine, as it may inflict unnecessary pain or discomfort on animals [40,70].

In scientific research aimed at identifying bacteria, the feces are typically frozen to −80 °C, and the DNA is extracted. Subsequently, the 16S rRNA gene is sequenced and analyzed using a computer program. The obtained sequences are then compared with existing databases of bacterial sequences [22,27,37]. It is important to note that these tests are not entirely precise, as they may not detect bacteria adhering to the mucous membrane, including enteroinvasive strains of *Escherichia coli* (EIEC) [15]. These components, along with genome analysis and the exploration of the relationship between the microbiotic environment of the gut and the host, constitute the focus of microbiome research [12].

### 5.3. Therapy and Prevention

In the context of the microbiome, the use of probiotics, prebiotics, and fecal microbiota transplantation (FMT) allows for achieving normobiosis, which, in turn, may contribute to reducing the genotoxicity of bacteria and halting the activation of inflammatory pathways. These approaches may positively influence the inhibition of carcinogenic processes [19,28]. However, opinions regarding the effectiveness of probiotics and prebiotics in animals still vary. Research on products based on various probiotic bacteria is not clear. The use of pre- and probiotics resulted in various types of effects compared to the groups treated with a placebo—from no difference in the effects of therapy, through an increase in proteins that can support therapy, to demonstrating a more effective clinical effect when used for a sufficiently long time [71,72]. Microbiota transplants also show promising results [73]. In the context of IBD, the FMT method has been demonstrated to confer notable benefits, including the alleviation of clinical symptoms and expansion of the intestinal microbiome towards beneficial bacteria. This is evidenced by findings from placebo studies. Additionally, it improves the physical properties of feces, even in cases unresponsive to previous treatment [74,75].

## 6. Conclusions

The microbiota’s impact on the development of lymphoma, particularly in veterinary medicine, remains underexplored. However, existing evidence does suggest that the composition of intestinal microorganisms plays a crucial role in maintaining a balanced and healthy immune system in animals, contributing to overall body homeostasis. Disruptions in this balance may lead to chronic inflammation and immune response disturbances. When combined with environmental and genetic factors, such disruptions can potentially contribute to the development and progression of cancer of the lymphatic system.

Although research on pre- and probiotics in veterinary oncology is still in its infancy, their potential to support the prevention and treatment of lymphoma is promising. Prebiotics and probiotics may help restore microbial balance, reduce inflammation, and modulate immune responses, providing a therapeutic avenue for the treatment of lymphoproliferative disorders and other microbiota-related diseases.

The hypothesized link between the microbial ecosystem and carcinogenesis encourages ongoing research in both veterinary and human medicine. As the understanding of microbial communities and lymphoma development grows, more insight is gained into their intricate relationship, paving the way for novel therapeutic interventions. An interdisciplinary approach, integrating molecular, environmental, and clinical perspectives, is crucial for elucidating the mechanisms underlying this pathology.

Isolating specific bacteria associated with intestinal lymphoma could serve as potential biomarkers for the disease. Identifying the stage at which microbial alterations emerge is essential for improving disease diagnosis, enabling timely treatment, and potentially implementing preventive measures if bacterial composition changes precede the onset of cancer.

## Figures and Tables

**Figure 1 cancers-16-02255-f001:**
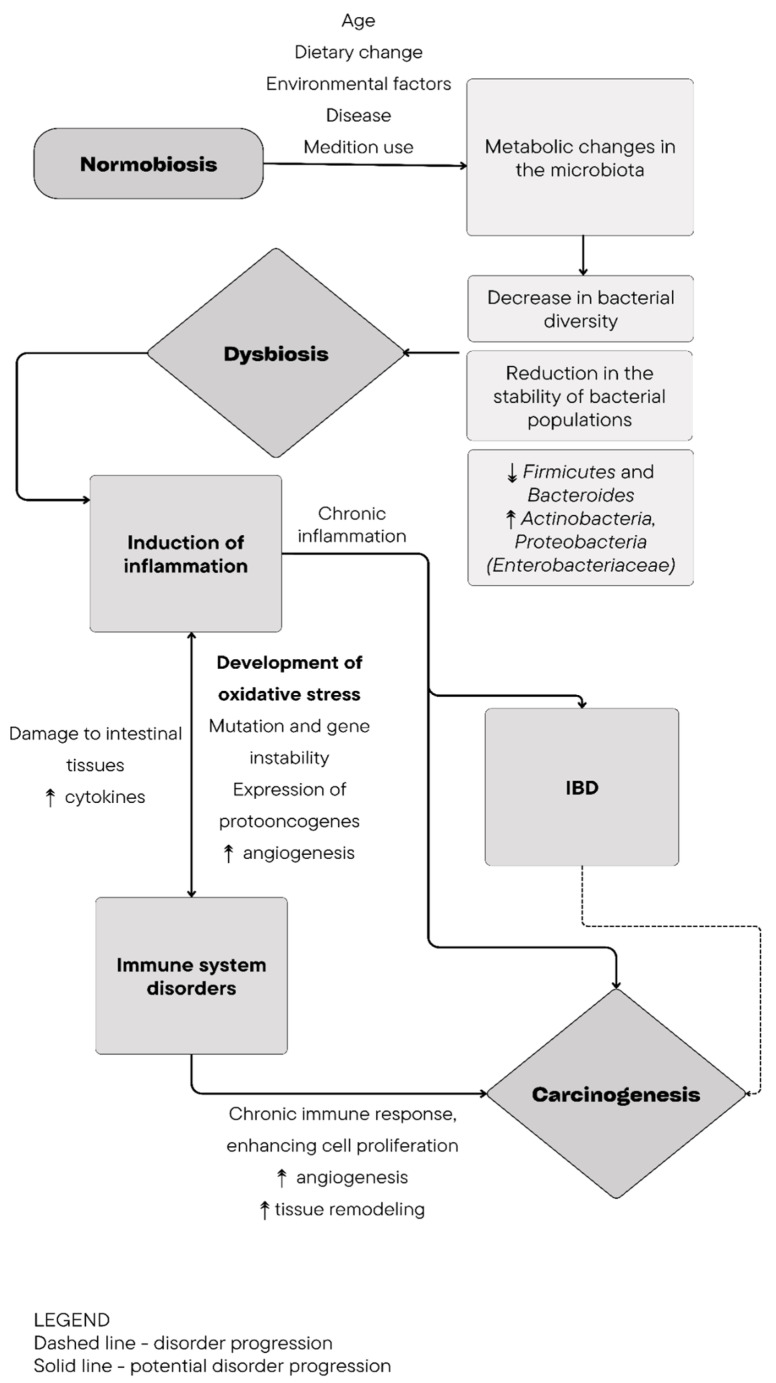
The origin of carcinogenesis associated with dysbiosis [9].

**Figure 2 cancers-16-02255-f002:**
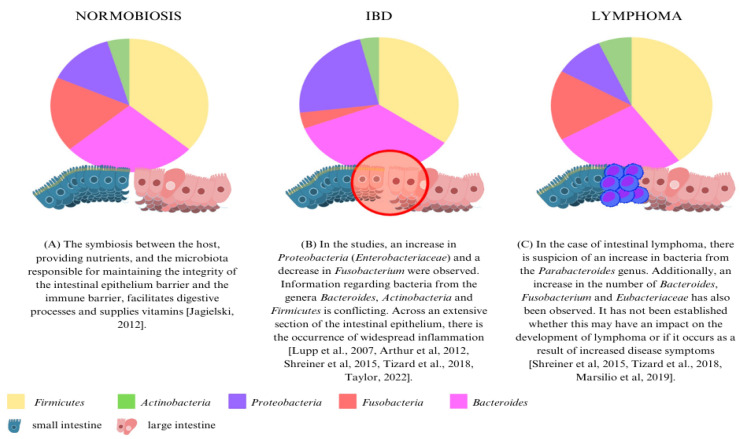
Comparison of microbiota in normobiosis, IBD and lymphoma. The pie charts illustrate the shift in proportions of different microbial populations in described states (the wedge sizes do not represent exact percentages) [2,21,35,36,37,38,39].

**Table 1 cancers-16-02255-t001:** Physiological microbiota of dogs and cats.

Group of Microorganisms *	Bacteria	Fungi	Viruses
Subject of Analysis			
Characteristics of a specific microbiome element	Intestinal microbiota changes depending on the section of the gastrointestinal tract. The diversity of bacteria increases along the gastrointestinal tract from the duodenum to the colon [21].	They are usually not found in the content of the intestines but adhere to the mucous membrane [13].	The quantity of viruses in the overall microbiota is small [13].
Dog	*Firmicutes*, *Bacteroidetes*, *Proteobacteria*, *Fusobacteria*, *Actinobacteria* [22,23]	Yeasts and molds Each dog has a unique profile of species of fungi present in their population [13].	Rotaviruses, coronaviruses, parvoviruses, noroviruses, astroviruses, paramyxoviruses [13]
Cat	*Firmicutes* (including *Clostridium* spp.), *Proteobacteria* (including *Enterobacteriaceae, Helicobacter*), *Bacteroidetes*, *Fusobacteria*, *Actinobacteria* [22,23]	*Ascomycota* (>90%); *Saccharomyces* and *Aspergillus*, *Neocallimastigomycota* (>5%) [24]	Rotaviruses, coronaviruses, parvoviruses, bacteriophages Caudovirales, Crenarchaeota, Euryarchaeota, Korarchaeota, Nanoarchaeota and Thaumarchaeota [3,24]

* ordered by frequency—from the most to the least common.

## Data Availability

No new data were created or analyzed in this study. Data sharing is not applicable to this article.

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
