# Peer review of "The Importance of Intestinal Microbiota and Dysbiosis in the Context of the Development of Intestinal Lymphoma in Dogs and Cats"

_cancers, 2024, doi:10.3390/cancers16122255_

Round 1

Reviewer 1 Report

Comments and Suggestions for Authors

Comments to Authors:

1.       Line 17-18: re-write the sentence, now it is difficult to read.

2.       Language seems too poor. Please check the article from journal language services.

3.       Abstract is too poor, difficult to read. Rewrite.

4.       Line 35: percentage value?

5.       Line 46-49: re-write the sentence.

6.       Figure 2 should be labelled properly.

7.       Provide statistic for lymphomas in dogs and cats.

8.       Give table providing studies on gut microbiota and lymphoma in animals.

9.       The theory of gut microbiota in relation with lymphoma in dogs and cats should be given.

Comments on the Quality of English Language

Too poor

Author Response

Dear Reviewer,

Please find enclosed the response in the attached file.

Yours sincerely,

Wioleta Breczko

Reviewer 2 Report

Comments and Suggestions for Authors

The authors reviewed the Importance of Intestinal Microbiota and Dysbiosis in the Context of the Development of Intestinal Lymphoma in Dogs and Cats.   However, the specific aim is not written clear in the Abstract and Introduction.     English language used is in spoken and vague language in many places of this manuscript. Need to be written more specific, clearer and more concise. Please see comments on English language below.   Figure 2, the composition of different bacteria in each subfigure isn't representative. Please think a better way to present.   

Comments on the Quality of English Language

English language used is in spoken and vague language in many places of this manuscript. Need to be written more specific, clearer and more concise. 

For example: Abstract line 21-28. This can also be the reason for an increased production and susceptibility of the organism to toxins and toxic metabolites produced by microorganisms. These factors contribute to the development of intestinal lymphoma. The theory of the microbiota's influence on the occurrence of this type of tumors is currently under investigation. This is important for an increasingly comprehensive understanding of the role of commensal microorganisms in the body. This is a important issue in veterinary and human medicine, where mainly Hodgkin's lymphoma and non-Hodgkin's lymphoma are recognized in this context. In both cases, understanding the causes of lymphoma formation can contribute to both better prevention and treatment of this disease

Introduction, line 34-36. this type of transformation in cats. In dogs, they

are diagnosed less frequently, but still with an important percentage [1,2,3]. Due to the high number of cases involving veterinary patients with this diagnosis in veterinary practices, etc. 

Line 50, the occurrence of this type of lymphoma is 

Please be specific and clearer, please check the whole manuscript and rewrite in more professional English language.

Author Response

(The authors gave the same response as above.)

Reviewer 3 Report

Comments and Suggestions for Authors

The paper is supposed to analyze the impact of the microbiota on the development of lymphoma in veterinary cases. This aspect is clearly mentioned in the title and conclusions, unfortunately, the entire text lacks detailed examples and veterinary research.

Repetitions in the text about the microbiota and lymphoma

In the submitted text, certain repetitions were noticed, especially in the context of the role of the microbiota in the immune system. Here are a few examples:

"The microbiota is essential for immunological reasons - a microbiota with the correct qualitative and quantitative composition is one of the immune mechanisms of the body" (lines 63-64)

This statement is repeated almost verbatim in the next paragraph:

"The immune barrier's previously mentioned role involves collaboration between the body and intestinal microbiota. From one perspective, commensal microorganisms assist in maintaining the body's homeostasis; conversely, their antigens are tolerated by the host's immune system. Additionally, the microbiota also plays a role in priming the immune system to recognize potential threats" (lines 79-83)

Here, the idea of the microbiota's role in maintaining homeostasis and priming the immune system is repeated.

"The microbiota plays a crucial role in the mechanisms of the body's specific immunity and constitutes a vital component of the body's defense mechanisms" (lines 70-71)

This statement is similar to the one at the beginning of the section on the immune system:

"The microbiota is essential for immunological reasons - a microbiota with the correct qualitative and quantitative composition is one of the immune mechanisms of the body" (lines 63-64)

Some sentences could be reworded to improve clarity and flow (e.g. the sentence starting at line 90).

There is a lack of information on the relationship between the composition of the microbiota and the risk of lymphoma in different animal species.

The article should provide a more detailed discussion of the mechanisms through which the microbiota can influence the pathogenesis of lymphoma in animals.

Author Response

(The authors gave the same response as above.)

Round 2

Reviewer 1 Report

Comments and Suggestions for Authors

Authors revised the manuscript appropriately, thus I recommend to accept the said manuscript.

Reviewer 3 Report

Comments and Suggestions for Authors

Accept in present form